# Mediastinal Teratoma with Nephroblastomatous Elements: Case Report, Literature Review, and Comparison with Maturing Fetal Glomerulogenic Zone/Definitive Zone Ratio and Nephrogenic Rests

**DOI:** 10.3390/ijms252212427

**Published:** 2024-11-19

**Authors:** Bader Alfawaz, Khaldoun Koujok, Gilgamesh Eamer, Consolato M. Sergi

**Affiliations:** 1Department of Pathology, Children’s Hospital of Eastern Ontario, University of Ottawa, Ottawa, ON K1H 8L1, Canada; balfawaz@toh.ca; 2Department of Radiology, Children’s Hospital of Eastern Ontario, University of Ottawa, Ottawa, ON K1H 8L1, Canada; kkoujok@cheo.on.ca; 3Department of Surgery, Children’s Hospital of Eastern Ontario, University of Ottawa, Ottawa, ON K1H 8L1, Canada; geamer@cheo.on.ca

**Keywords:** Wilms’ tumor, Teratoid Wilms’ tumor, Nephroblastoma, immunohistochemistry, narrative review

## Abstract

Extrarenal teratoid Wilms’ tumor (TWT) is a variant of Wilms’ tumor with fewer than 30 cases reported in the literature. It comprises more than 50% heterologous tissue and presents a significant diagnostic challenge due to its complex histology. We report an unusual case of mediastinal teratoma with nephroblastomatous elements in an 8-year-old female. The patient presented with respiratory distress, fever, weight loss, and a large anterior mediastinal mass. Imaging revealed a heterogeneous tumor containing fat, fluid, and calcification, suggestive of a teratoma. Surgical resection confirmed a mature cystic teratoma with foci of nephroblastoma. Pathological analysis demonstrated a mixture of ectodermal, mesodermal, and endodermal tissues alongside nephroblastomatous components. Immunohistochemistry was positive for Wilms Tumor 1 and other relevant markers, confirming the diagnosis. The patient had an uneventful postoperative course and was discharged after three days. This case adds to the growing body of research on extrarenal TWT, particularly its occurrence in the mediastinum, a rare site for such tumors. A literature review highlighted that extrarenal TWT often affects children, typically presenting in the retroperitoneum or sacrococcygeal regions, with varying recurrence rates and long-term outcomes. This case underscores the importance of histopathological and immunohistochemical analysis in diagnosing TWT and differentiating it from other mediastinal tumors to ensure appropriate treatment planning, emphasizing the need for long-term follow-up due to the potential for recurrence or metastasis. This paper also provides an in-depth look at nephron development and nephrogenic rests, highlighting the structural and functional aspects of nephrogenesis and the factors that disrupt it in fetal kidneys.

## 1. Introduction

Extrarenal Wilms’ tumor is an extremely rare neoplasm, with fewer than 100 documented cases, that often presents in the retroperitoneum, sacrococcygeal region, mediastinum, and other atypical extrarenal sites [1,2]. Teratoid Wilms’ tumor (TWT), a variant characterized by the presence of over 50% heterologous tissue and a component of immature renal elements, remains a diagnostic challenge due to its rarity and complex histology [3]. There are less than 30 cases of extrarenal TWT reported in the literature. Here, we report a case of mediastinal teratoma with nephroblastomatous elements in an 8-year-old girl.

## 2. Case Presentation 

### 2.1. Clinical History 

An 8-year-old female that had recently immigrated from Hong Kong and was previously healthy presented to the emergency department of the Children’s Hospital of Eastern Ontario in Ottawa (ON, Canada) with a 3-week history of daily fever (37.5–39 °C), intermittent abdominal pain, and weight loss of 2–4 kg. Her obstetric history was unremarkable. Physical examination revealed tachypnea (respiratory rate: 30/min), tachycardia (heart rate: 125 bpm), and reduced air entry to the right chest. 

### 2.2. Radiology 

An initial chest X-ray demonstrated a large circumscribed dense opacity projecting over the right hemithorax with a hilum overlay sign, indicating a large anterior mediastinal mass. A chest CT with contrast showed a 7.0 × 7.5 × 9.8 cm circumscribed heterogenous partially enhancing mass with variably attenuating components, including adipose tissue, enhancing soft tissue, and dense calcific densities. No mediastinal or axillary lymphadenopathy was observed. The mass effects included compression of the right atrium and superior vena cava, a mild right-to-left shift of the heart, and right lung atelectasis (Figure 1a–d). An ultrasound of the abdomen and pelvis showed no abdominal or pelvic pathology. The overall impression was mediastinal teratoma. 

### 2.3. Biochemistry 

Blood work indicated leukocytosis (13.4 × 10^9^/L); thrombocytosis (551 × 10^9^/L); elevated C-reactive protein (59.8 mg/L); elevated EST (87 mm/h); and unremarkable creatinine, electrolytes, urinalysis, and liver function. Beta HCG and alpha-fetoprotein tumor markers were both negative. 

### 2.4. Surgical Management

Surgical resection via a sternotomy was performed. The intraoperative findings included a large intact right-sided intrathoracic mass extending over the midline and bordered by the superior vena cava superiorly and costal margin inferiorly. The lesion adhered to the pericardium without apparent invasion. Cyst contents and necrotic material were spilled within the right hemithorax and mediastinum during dissection. The entire mass and the fluid were submitted to pathology for evaluation. 

### 2.5. Pathology 

The specimen received in pathology consisted of a 10 × 8.0 × 2.5 cm cystic lesion attached to 6.0 × 2.0 × 1.0 cm of unremarkable thymic tissue. Cut sections showed predominantly cystic variegated and heterogeneous surfaces with mucus, cartilage, skin, and hair tissue. No grossly identifiable necrosis or suspicious nodules were seen. One representative section of the thymus and twelve representative sections of the cystic lesion were submitted. 

Microscopy showed a mature cystic teratoma with benign elements of ectodermal, mesodermal, and endodermal lineages. One focus spanning 1.3 cm had atypical epithelial tissue with a serpentine growth pattern exhibiting mostly blastemal differentiation, focal glomeruloid differentiation, tubular differentiation, frequent apoptotic bodies, and brisk mitotic activity. This tissue was, in places, interdigitated between unremarkable fibroadipose tissue (Figure 2a). Immunohistochemistry was positive for the Wilms Tumor 1 (WT1) marker, pan-cytokeratins, PAX8, CD56, SALL-4, and glypican-3 and negative for Oct3/4, CD30, and beta-HCG. The Ki-67 proliferation index was variable (Figure 2b–d). 

Table 1 provides a comprehensive list of all immunohistochemical antibodies used in this case study.

The thymus gland showed a normal architecture with the cortex and medulla containing Hassall’s corpuscles.

The differential diagnosis included teratoma with immature neuroepithelial elements and mixed germ cell tumor (i.e., yolk sac tumor, dysgerminoma, or embryonic carcinoma).

The final diagnosis was reported as cystic teratoma with unifocal nephroblastoma.

### 2.6. Postoperative Course

The patient had an uncomplicated postoperative stay and was discharged after three days.

## 3. Discussion

Here, we report on an unusual mediastinal teratoma that included three layers of embryonal differentiation. It included elements derived from ectodermal, mesodermal, and endodermal lineages and additional nephroblastomatous elements. In the following paragraphs, we will report on a literature review of extrarenal teratoid Wilms’ tumor described using the PubMed, Scopus, and Cochrane databases; delineate the developing glomerulogenic zone/definitive zone during fetal development; and comment on nephrogenic rests and their evolution. The search keywords included “teratoid Wilms’ tumor”, “teratoma with nephroblastomatous elements”, “teratoma with Wilms’ tumor”, “teratoma with immature renal elements”, and “teratoma with nephroblastoma”.

### 3.1. Narrative Literature Review

A review of the literature revealed 24 cases (including our case) of extrarenal TWT exhibiting either nephroblastomatous elements admixed with the teratoma or compact tissue with a “mass-effect” in patients with no concurrent or previous history of Wilms’ tumor, which are summarized in Table 2 [4,5,6,7,8,9,10,11,12,13,14,15,16,17,18,19,20,21,22,23,24]. The tumor sites were variable, with eight (33%) cases observed within the retroperitoneum. The remaining sites were in the sacrococcygeal region (25%, six cases), testes (12.5%, three cases), mediastinum (12.5%, three cases), abdomen (4%, one case), uterus (4%, one case), and vagina (4%, one case). The median age was 3 years (newborn–62 years). There was a slight female predilection, with a female-to-male ratio of 1.7:1 (15 vs. 9 cases, respectively). The median tumor size was 9.0 cm (2.0–20.0 cm). Fifteen patients had no documented recurrence, while six had documented recurrences. Three patients had documented deaths from disease, and one patient died from other causes. The median follow-up duration of the cases with no recurrence was 14 months (3–85 months). 

TWT is a rare histological variant of Wilms’ tumor. The nomenclature of such tumors is yet to be standardized, with terms such as teratoma with nephroblastoma, teratoma with nephroblastomatous elements, and teratoma with immature renal elements being used in the literature to describe similar entities. It is characterized by heterologous elements, typically comprising more than 50% of the tumor mass. These elements can include tissues derived from all three germ layers, such as squamous epithelium; adipose tissue; skeletal muscle; and, occasionally, neuroglial tissue. TWT was first described by Variend et al. in 1984. They recognized the unusual presence of teratoid elements within Wilms’ tumors. TWT is exceedingly rare, with fewer than 30 cases reported in the literature. These tumors typically present in children aged 1 to 10 years and are often located in the kidneys, though extrarenal cases have been documented in the retroperitoneum, the sacrococcygeal region, and other sites [3,25,26].

Mediastinal TWTs are exceptionally uncommon, with very few documented cases. In these instances, the tumors present similarly to other mediastinal masses, often leading to misdiagnoses. The reported cases primarily involve young children, with a slight female predominance. Given the tumor’s rare occurrence in the mediastinum, demographic data remain limited, though these cases typically affect patients under five years of age [13,27].

The pathogenesis of TWT is still a subject of research and debate. It is believed that TWT may originate from nephrogenic rests that have the potential to differentiate into a wide variety of tissues, including those not typically found in Wilms’ tumor. This differentiation leads to the formation of teratoid elements, which are less responsive to conventional therapies like chemotherapy and radiation. Some researchers hypothesize that these tumors may represent an extreme heterologous differentiation rather than a separate entity [28,29].

Diagnosing TWT is challenging due to its rarity and the diverse histological components it contains. Histopathological examination is crucial, as it identifies triphasic nephroblastic elements (blastemal, stromal, and epithelial) and the essential teratoid components for diagnosis. Immunohistochemical testing plays a significant role in confirming a diagnosis of TWT, especially when a tumor presents with a diverse or atypical histology. Markers such as WT1, which is typically positive in Wilms’ tumor (in addition to PAX8, CD56, and CD57), help confirm a nephroblastic origin, while others like cytokeratins, desmin, and S-100 protein assist in identifying the specific tissue types within a tumor [17,28,29].

### 3.2. Glomerulogenic Zone During Ontogenesis

During fetal development and ontogenesis, the kidneys show variation in their glomerulogenic zone to definitive zone ratio, which is characteristic of gestational age (Figure 3). 

Various noxae with distinct molecular compositions might induce premature cessation of nephron development in preterm and low birth-weight infants [30,31,32]. Despite extensive analysis of clinical parameters, there is a paucity of information concerning the impacted stages of nephron anlage, the specific cellular targets of noxae, and the molecular mechanisms that result in pathological changes in the outer cortex of the fetal human kidney. This dilemma is exacerbated by the need for meticulous microscopic evaluation of specimens, as the target area of noxae is intricately structured. Enveloped by the renal capsule, it comprises fetal, maturing, and adult tissue layers [33]. This indicates a structural gradient wherein nephron development is integrated. This process is initiated adjacent to the inner aspect of the renal capsule, extending perpendicularly into the external nephrogenic zone, the underlying maturation zone, and subsequently into the mature zone. 

The temporarily present stages of nephron anlage, which are confined to the nephrogenic zone, are significant for the disruption of nephrogenesis. The nephrogenic niche encompasses the progressively forming pretubular aggregate, renal vesicles, and comma- and S-shaped structures. Utilizing the mouse kidney as an experimental model, it was shown that exposure to noxae affects the nephrogenic niche, leading to diminished expression of morphogenic molecules, including Gfrα1, Gdnf, Bmp4, Pax2, and Six2 [34]. Remarkably, comparable evidence about injuries during the early stages of nephron anlage in the unborn human kidney is deficient. The available evidence indicates that the breadth of the nephrogenic zone is 150 µm in gestational controls. However, in preterm infants, it is reduced considerably to 100 µm [35]. Moreover, the absence of basophilic S-shaped structures in the nephrogenic zone of preterm infants has been documented [36]. Nevertheless, these papers do not include information regarding the pathological changes in the tissues surrounding the first nephron. 

Atypical glomeruli, characterized by a dilated Bowman’s space, a reduced glomerular tuft, and a decrease in the number of glomeruli, are frequently seen. Nonetheless, these pathological symptoms mostly pertain to the maturing nephron in the underlying maturation zone rather than the stages of nephron anlage in the nephrogenic zone. Unexpectedly, no prior relevant papers offer definitive evidence about whether noxae that affect nephrogenesis directly influence the growing nephron or if their impact is indirect, resulting from disruptions in the surrounding tissues and associated molecular interactions. 

Studying the relevant literature unexpectedly showed the nephrogenic zone’s significance during nephron formation inside the fetal human kidney and that its status as the principal target for agents disrupting nephrogenesis has seldom been investigated. In this context, it is comprehensible that the delineated stages of nephron anlage and nephron formation have only been presented in recent years as initial systematic morphological data addressing the characteristics of the nephrogenic zone [37,38,39]. The data imply that each stage of nephron anlage undergoes distinct placement, orientation, and sculpting. An additional significant observation in this context is that throughout development, both the individual stage of nephron anlage and the associated covering tissues undergo particular alterations. This morphological study aims to identify the exact location of nephron development, provide essential coordinates, and record the interrelationship of its surrounding tissues. 

Noxae that hinder nephrogenesis affect the outer cortex of the fetal human kidney during late gestation. Enveloped by the renal capsule, it comprises transverse layers of embryonic, developing, and fully mature parenchyma and stroma. Nephron production commences in the exterior nephrogenic zone, identifiable by the ephemeral phases of nephron anlage. In the underlying maturation zone, the transformation of the S-shaped structure, as the final phase of nephron anlage, into the permanent nephron takes place. This process involves elongation, geographical expansion, and functional specialization of individual nephron segments. The definitive nephron’s characteristic morphological traits are established in the underlying developed zone. 

The exterior nephrogenic zone is most pertinent when investigating early impressions resulting from nephrogenic dysfunction. This is observable beneath the renal capsule as a slender band of embryonic parenchyma and stroma [40]. The outside border of the fetal human kidney is next to the inner surface of the renal capsule. The inner border was previously characterized as a “dotted” line intersecting the proximal (medulla-oriented) pole of the laterally aligned S-shaped structures in a transverse orientation [41]. Moreover, it has been established that the vertical distance from the renal capsule to the proximal pole of an S-shaped structure is 150 µm. The nephrogenic zone’s inner boundary adjoins the underlying maturation zone. The nephrogenic zone comprises nephrogenic and stromal progenitor cells and temporary stages of nephron anlage, including the nephrogenic niche, pretubular aggregation, renal vesicles, and comma- and S-shaped structures. Furthermore, the ampullae of the collecting ducts produced from the ureteric bud elongate at the distal termini of the collecting duct tubules. The perforating radiate arteries align vertically. Numerous macrophages are observable in the emerging interstitium and throughout the microvascular system [42]. 

### 3.3. Nephrogenic Rests

A critical consideration in this process is distinguishing between nephrogenic rests (NRs) and early nephroblastoma, as NRs are believed to be precursors to Wilms’ tumors (Figure 4). Within the kidneys, NRs are classified into two major categories: perilobar (PLNRs) and intralobar (ILNRs). Further subclassifications based on their morphologies are used to classify them into dormant, maturing, sclerosing, hyperplastic, and neoplastic rests. Differentiation between these subtypes, particularly distinguishing hyperplastic rests from neoplastic rests, is vital, as neoplastic rests exhibit compression of adjacent tissues, forming densely cellular and expansile nodules that may resemble early Wilms’ tumors. 

Neoplastic nodules demonstrating more crowding and considerable mitotic activity are best classified as nephroblastomatous rests or Wilms’ tumors [43]. A diagnosis of Wilms’ tumor within the kidney typically relies on well-established histological criteria that may not be fully applicable to diagnosing teratoid Wilms’ tumor. In our case, the presence of prominent mitotic activity within the premature renal elements, in the absence of nodule formation, was sufficient to consider the diagnosis of TWT. 

The term nephrogenic rest (NR) is used for any lesions considered probable antecedents of Wilms’ tumors. The phrase embryonal remainder denotes clusters of embryonal cells or tissues that endure throughout postnatal existence and maintain the capacity to generate embryonic cancers [44]. The phrase “nephrogenic rest” effectively communicates the idea of a pause that emerged during nephrogenesis, and yet it is not excessively limiting. The widely used term persistent nodular blastema is not an accurate descriptor. Some have utilized the term nephroblastomatosis as a synonym for nephrogenic rest, but they are different concepts. A nephrogenic rest is a locus of aberrantly persistent nephrogenic cells that can be stimulated to develop a Wilms’ tumor, while nephroblastomatosis is the widespread or multifocal occurrence of nephrogenic rests or their acknowledged derivatives. This term also applies in instances where remnants’ earlier presence can be deduced (e.g., multicentric and bilateral tumors). 

As indicated above, the renal lobe constitutes the fundamental organizational unit of the metanephros. Each lobe comprises a medullary pyramid and its cortical mantle. The renal lobule, frequently conflated with the lobe, is a diminutive cortical region centered on a solitary medullary ray. A basic link exists between lobar topography and the timeline of renal development. The emergence of collecting duct precursors triggers lobar organization, making the medullary pyramid the first segment of the renal lobe to manifest. Upon establishing a fundamental medullary template, nephrons are incrementally added to its periphery over roughly 12 generations. Generally, the nephrons closest to the lobar surface are the youngest, while the oldest nephrons are situated further away in the medulla, as shown in the previous microphotograph. 

This correlation between lobar topography and developmental chronology offers crucial indicators regarding the timing of teratogenic events in the kidney. Developmental abnormalities occurring late in embryonic development often manifest mainly in the peripheral lobes. Earlier developmental disruptions will frequently occur deeper inside the brain or medulla, and abnormalities at a very early stage can cause complete disruption of lobar organization. Nephrons were initially seen in the periphery of the renal lobe, suggesting a comparatively late developmental aberration characterized by a “teratogenic termination period” [45] before the termination of nephrogenesis, typically around the 36th week of gestation. The evolution from a blastema to a Wilms’ tumor is illustrated in Figure 4.

NRs manifest in unifocal, multifocal, and diffuse distribution patterns. Focal PLNRs typically have a rounded or oval morphology. A diffuse PLNR creates a continuous band-like region at the lobar margin. ILNRs are occasionally multifocal but are infrequently diffuse. Diffuse ILNRs are equivalent to universal nephroblastomatosis, as detailed below. The categories of ILNRs and PLNRs each include a variety of morphological forms. Rests differ in their dimensions, morphologies, distributions, cellular compositions, and degrees of differentiation, along with the presence or absence of overlaid neoplasm development. Rests may also exhibit two unique types of proliferative alterations. We may encounter (a) generalized growth of the remainder as a whole (hyperplastic NRs), or (b) localized cellular proliferation (probably clonal) inside the remainder (neoplastic NRs). Neoplastic NRs may encompass benign-appearing (adenomatous) tumors. Maturation and sclerosis frequently transpire in hyperplastic remnants.

The essential distinction between the hyperplastic and neoplastic forms of rest overgrowth exists, but it is quite vague. Their differentiation is often ambiguous. Hyperplastic rests maintain the forms of the original rests and do not include squeezed residual pieces near the periphery. Hyperplastic rests may occasionally attain considerable sizes. However, there is no evidence indicating that they can metastasize. Nonetheless, they might increase the likelihood of neoplastic induction by elevating the quantity of vulnerable cells. Neoplastic rests are defined by swelling, typically spherical nodules. Compressed fragments of the original rests are frequently apparent at the margins of these tiny tumors but are ultimately obliterated by tumor expansion. Distinctions between adenomatous and nephroblastomatous neoplasms are much more complex. Nascent or dormant PLNRs are tiny remnants primarily composed of blastemal cells. Such rests are typically recognized in infants; nevertheless, we have also observed them in older children and teenagers. Mitotic figures are infrequently observed. These minute components are susceptible to harm and can be obliterated by the removal of the renal capsule, a surgery with minimal applicability in managing pediatric renal specimens. Involuting and obsolete PLNRs exhibit varying degrees of epithelial and stromal differentiation and stromal sclerosis. Tubular structures are bordered by a monolayer of low cuboidal, basophilic epithelial cells encircled by variably hyalinized collagenous tissue. Psammoma bodies are frequently observed, along with aberrant glomeruli. Involuting PLNRs are associated with the “sclerosing metanephric hamartomatous background” [43,46,47,48,49,50,51]. The presence of mostly spherical formations distinguishes neoplastic PLNRs. Growing nodules of highly aggregated cells within latent, hyperplastic, or sclerosing remnants may have a tendency to infiltrate. However, the cellular density of the tumor is typically elevated compared to the adjacent tissue, with variability among lesions. There are tightly clustered blastemal or embryonal epithelial components that exhibit multiple mitotic figures, and they are indistinguishable from ordinary Wilms’ tumors. They establish a continuum ranging from tiny dimensions to substantial, clinically evident masses. The “two-hit” hypothesis proposed by Knudson and others [52,53,54] has long provided a valuable conceptual framework for elucidating the pathophysiologies of specific embryonal cancers, including Wilms’ tumors. 

## 4. In-Depth Analysis of Nephrogenic Rest–Wilms’ Tumor Nexus

The nexus of the glomerulogenic zone, nephrogenic rests (NRs), and Wilms’ tumor has been extensively examined and elucidated based on our present understanding. Further research will be necessary in the future, and laser-capture microdissection is likely to play a crucial role soon. As indicated above, NRs are significantly persistent clusters of embryonic cells indicative of microscopic abnormalities (dysplasias) in the developing kidney. Although NRs are primarily recognized as precursors to Wilms’ tumor, numerous alternate outcomes are evident, with the majority of rests ultimately leading to atresia. NRs exhibit a distinct boundary with the surrounding kidney and do not possess a pseudocapsule, while Wilms’ tumors consistently display a pseudocapsule that separates the tumor from the underlying kidney. Nephroblastomatosis refers to multifocal or diffuse nephrogenic rests, indicating a more widespread disease state. Universal nephroblastomatosis signifies complete substitution of the renal lobe with nephrogenic tissue. The outcomes of nephrogenic rests and nephroblastomatosis include regression, sclerosis, quiescence, hyperplasia, and neoplasia. Evidence strongly indicates that the neoplastic transformation of nephrogenic rests leads to nephroblastoma. NRs predominantly arise in conjunction with Wilms’ tumor; perilobar rests exhibit a significant correlation with synchronous bilateral Wilms’ tumors, while intralobar rests are more closely linked to metachronous tumors. Genetic research indicates that NRs frequently exhibit numerous chromosomal abnormalities similar to those found in Wilms’ tumors, reinforcing the notion that they are progenitors to nephroblastoma. Consequently, NRs are acknowledged as therapeutically important entities that need thorough diagnoses and vigilant monitoring. Increased awareness of the clinical significance of NRs and nephroblastomatosis has resulted in enhanced detection of these precancerous lesions, stimulated more rigorous research into their biological behavior, and prompted extensive discussions regarding potential new treatment protocols. Pathologists’ capacity to identify and detect NRs has improved due to the enhanced Beckwith categorization [55]. Radiologists have utilized high-resolution imaging methods to enhance the detection of NRs in situ. Clinicians and surgeons are more cognizant of the influence that NRs exert on patient treatment. Notwithstanding this advancement, additional evidence is required to more precisely characterize these lesions. An NR that may subsequently evolve into a Wilms’ tumor could emerge at any phase of glomerulogenic zone differentiation. However, the latest insights into kidney development indicate that these rests likely originate from the initial stages of embryonic or fetal development, when the glomerulogenic zone is more pronounced compared to the later stages of gestation. Bánki et al. [56] provided a comprehensive review of all molecular investigations on nephrogenic rests conducted between 1990 and 2022. Consequently, an extensive array of approaches was employed to investigate chromosomal areas, copy number variations, specific genes, and epigenetic alterations in NRs. Twenty-three papers were identified, demonstrating the loss of chromosomal arms 11p13, 11p15, 1p, 4q, and 11p, alongside gains in 1q, 7q, and 12q, in addition to alterations in *WT1* (Table 3). 

Consequently, significant insights have been gained into the genomic alterations in NRs. Loss of heterozygosity at 11p13 and 11p15, along with the expression of IGF-2 and mutations in *WT1* and *WTX*, seems to contribute to the early carcinogenesis of Wilms’ tumor. Loss of heterozygosity at 16q and 7p, along with mutations in *CTNNB1*, appears to manifest later in development. The methylation patterns of NRs are comparable to those of WTs. Owing to swift advancements in genome-wide molecular techniques, enhanced utilization of formalin-fixed and paraffin-embedded materials, and the accessibility of histologically validated frozen specimens, genetic alterations in NRs and corresponding Wilms’ tumors can be examined in larger cohorts, potentially elucidating the initial stages of tumorigenesis of the nephroblastoma. Single-cell transcriptomics associated with laser-capture microdissection will probably be crucial in the near future.

## 5. Conclusions

Management of TWTs primarily involves surgical resection, as these tumors often show resistance to chemotherapy, mainly due to the presence of mature, differentiated tissues. The prognosis for patients with TWTs is generally favorable if their tumors are completely excised, although long-term follow-up is necessary due to the risk of recurrence or metastasis [27]. These diagnostic considerations are essential for distinguishing TWTs from other mediastinal neoplasms to ensure accurate diagnosis and appropriate treatment planning.

## Figures and Tables

**Figure 1 ijms-25-12427-f001:**
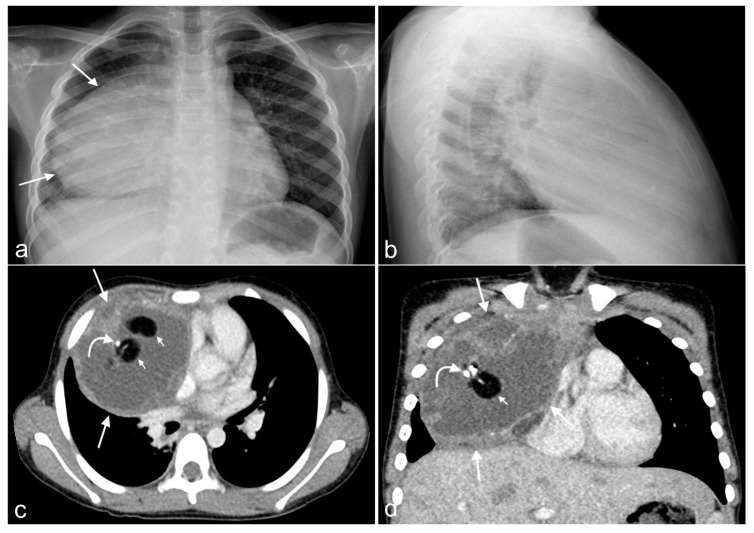
Imaging. (**a**) This PA view of the chest demonstrates a large mass on the right side of the chest (arrows). (**b**) A lateral view shows that the mass in the anterior mediastinum overlaps with the heart shadow. Axial (**c**) and coronal (**d**) contrast-enhanced CT images demonstrate an anterior mediastinal tumor (large arrows) causing mass effects in the heart and great vessels. It contains fat (small arrows), fluid, and calcification (curved arrow). The location and the presence of fat, fluid, and calcification are characteristic of a teratoma.

**Figure 2 ijms-25-12427-f002:**
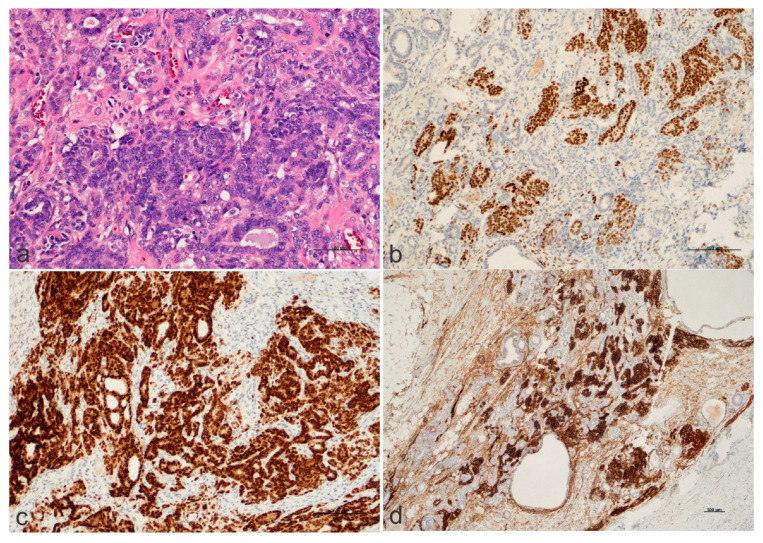
**The histology and immunohistochemistry of the mediastinal tumor.** This figure includes four microphotographs with included scale bars, i.e., (**a**) hematoxylin and eosin staining (200× magnification, scale bar 50 μm); (**b**) WT1 expression (100× magnification, scale bar 100 μm); (**c**) PAX8 expression (100× magnification, scale bar 100 μm); and (**d**) expression of CD56, which is a sensitive neuroendocrine marker (40× magnification, scale bar 100 μm). *PAX8* is a gene that encodes a transcription factor involved in the development of the thyroid, renal, and Müllerian systems.

**Figure 3 ijms-25-12427-f003:**
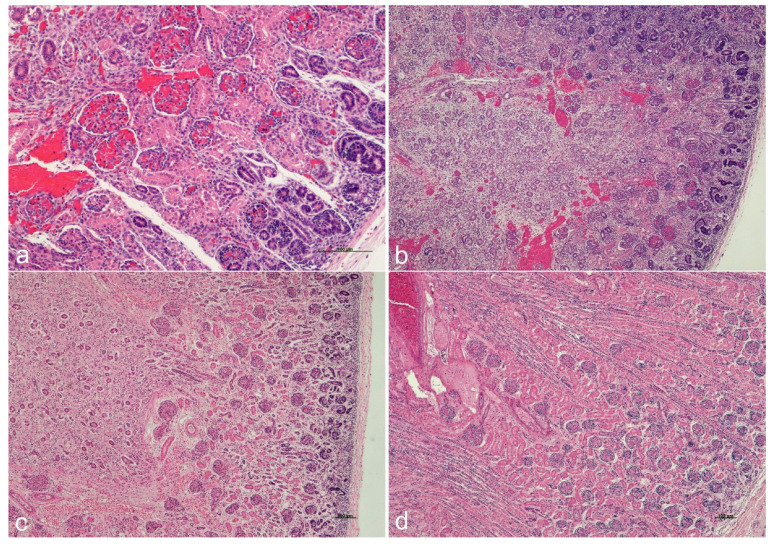
**The glomerulogenic zone during ontogenesis.** The glomerulogenic zone varies during ontogenesis with a decrease in thickness as the pregnancy reaches full term. The ratio of the glomerulogenic zone (GZ) to the definitive zone (DZ) is particularly critical because it mirrors the maturation of the renal parenchyma. GZ/DZ at 17 + 2 weeks of gestation (**a**), 20 + 1 weeks of gestation (**b**), 22 + 6 weeks of gestation (**c**), and 39 + 6 weeks of gestation (**d**) (hematoxylin and eosin staining). Scale bars (100 μm) are embedded in the microphotographs a, c, and d. The microphotograph b was taken at the same magnification as c. Thus, the scale bar used for the microphotograph c can be used for the microphotograph b. All microphotographs were taken during clinical autopsies performed after spontaneous stillbirths without evidence of maturation delay, and no congenital anomalies were identified during the autopsies. In all cases, consent for autopsy was granted without restrictions, and there was permission to use microphotographs for academic and educational purposes.

**Figure 4 ijms-25-12427-f004:**
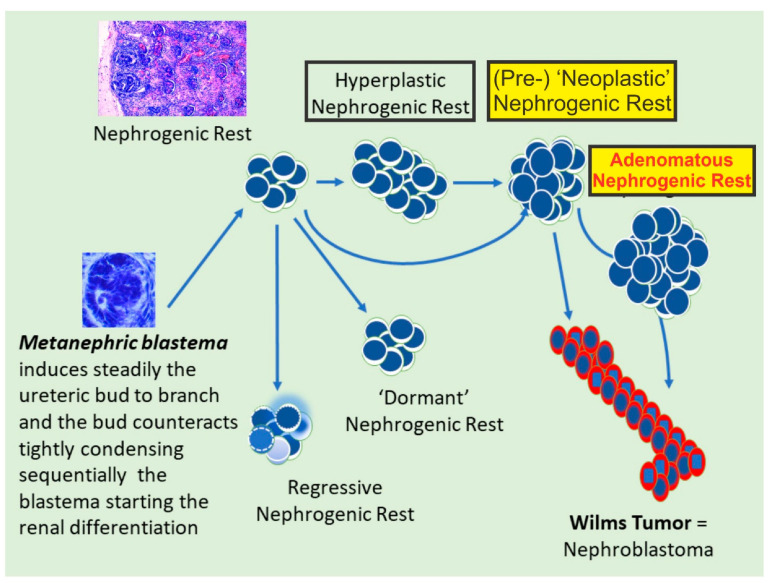
Schematic reproduction of a metanephric blastema and its evolution. A metanephric blastema steadily induces the ureteric bud to branch. The bud counteracts, tightly condensing. Sequentially, the blastema initiates renal differentiation. Potential sequelae are also displayed (see the text for details).

**Table 1 ijms-25-12427-t001:** A list of the antibodies used for the immunohistochemistry studies.

Antibody	Vendor	Catalogue	Clone	ER	Time (min)	Dilution
Pan-Keratins	In-house mix of CK8/18 and AE 1/3 (see below)			ER1	20	1:750
CK8/18	Leica	PA0067	5D3	ER1	20	RTU *
AE 1/3	Millipore Sigma	MAB2314	AE1–3	ER1	20	1:750
CD56	Leica	PA0191	CD564	ER2	15	RTU *
Glypican-3	Millipore Sigma	CMQ-261M96	-	ER2	20	1:50
Beta-HCG	Leica	PA0014	POLY **	ER1	20	RTU *
Ki67	Dako	M7240	MIB-1	ER1	20	1:75
Oct 3/4	Leica	PA0193	N1NK	ER2	30	RTU *
PAX8	Millipore Sigma	363M-14	MRQ-50	ER2	20	1:100
SALL4	Biocare	CM384C	6 × 10^3^	ER2	20	1:100
Wilms Tumor 1	Leica	PA0562	WT49	ER2	40	RTU *

Notes: ER, Epitope Retrieval; ER1, Epitope Retrieval Solution 1 (Leica); ER2, Epitope Retrieval Solution 2 (Leica). * BOND ready to use (RTU); ** Polyclonal Beta-HCG antibody from Leica.

**Table 2 ijms-25-12427-t002:** Reported cases of extrarenal teratoid Wilms’ tumor (TWT).

Author	YR	Age	Sex	Site	Size	Histology	Chemo/Rad	Follow-Up
Moyson [4]	1961	3 y	F	Mediastinum	“Newborn head”			Death (several weeks later)
Malik [5]	1967	6 y	F	Retroperitoneum	12.0 × 12.0 × 8.0 cm			N.A.
Tebbi [6]	1974	3 y	F	Sacrococcygeal region	8.0 × 6.0 × 6.0 cm	Triphasic WT, cysts with mucinous epithelium, smooth muscle, fat, and cartilage	5400 rads over 6 weeks Vincristine, actinomycin D for 12 weeks	No recurrence (46 months)
Carney [7]	1975	41 y	M	Retroperitoneum	Not specified	Triphasic WT, renal cell carcinoma, teratoma elements	Radiation (2 months after surgery)	Death (4 months)
Valdiserri [8]	1981	3 d	F	Sacrococcygeal region	9.0 cm	Immature teratoma with predominant renal immature component	None	No recurrence (6 years)
Valdiserri [8]	1981	1.5 y	F	Sacrococcygeal region	2.0 cm	Immature teratoma with predominant renal immature component	None	Death (3 years, after surgery for other causes)
Kim [9]	1990	10 m	F	Retroperitoneum	3.0 × 3.0 × 2.5 cm	Predominant WT elements, minor component of cartilage, adipose tissue, glial tissue, mucinous glands	Vincristine, cyclophosphamide, and actinomycin-D	No recurrence (4 months)
Park [10]	1991	4 y	F	Retroperitoneum	20.0 × 15.0 × 12.0 cm	Predominant heterologous elements with adipose tissue, cartilage, ganglia, skeletal muscle, and glandular structures. Minor component of WT	Refused chemotherapy	Recurrence (3 months later with only nephroblastomatous elements)
Pawel [11]	1998	7 y	M	Supernumerary ectopic ureteropelvic structure	5.0 cm	Botryoid RMS-like growth with areas of triphasic WT	Vincristine and dactinomycin (18 weeks)	No recurrence (18 months)
Emerson [12]	2004	22 y	M	Left testis with left supraclavicular and retroperitoneal lymphadenopathy	7.5 cm	Mature and immature teratoma with WT and RMS. Supraclavicular lymph node with mature teratoma	Bleomycin–etopside–cisplatin (3 cycles)	No recurrence (21 months)
Badillo [13]	2006	18 y	M	Mediastinum	16.0 × 13.0 × 7.0 cm	Nodular triphasic WT and mature heterologous background elements	Vincristine and actinomycin D (after recurrence)	Recurrence
Vanasupa [14]	2007	18 y	M	Right testis	3.0 cm	Teratoma with WT, RMS, focal embryonal carcinoma, and minute YST	None	No recurrence (4 months)
García-Galvis [15]	2008	62 y	F	Uterus	8.0 cm	Triphasic WT and abundant heterologous elements	Cisplatin and ifosfamide (4 cycles) followed by 50 Gy radiation therapy	No recurrence (14 months)
Al-Adnani [16]	2009	15 m	F	Sacrococcygeal region	9.0 × 7.0 × 4.5 cm	Mature tissues and “large areas of nephrogenic differentiation”	None	No recurrence (9 months)
Song [17]	2010	13 y	F	Vagina	6.0 × 5.0 cm	A diffuse and nodular “blastema”, rhabdomyomatous spindle cells, fetal glomeruli in a mature cystic teratoma	Vincristine, cyclophosphamide, and actinomycin-D for 6 months	No recurrence (7 years, 1 month)
Song [17]	2010	1 d	F	Sacrococcygeal region	13 cm	A diffuse and nodular “blastema”, rhabdomyomatous spindle cells, fetal glomeruli in a mature cystic teratoma	Vincristine and actinomycin-D for 6 months	No recurrence (2 years, 5 months)
Chowhan [18]	2011	15 m	M	Retroperitoneum	6.0 × 6.0 cm	“Heterologous elements”, focally dilated glandular structures, focal adipocytes, neural and skeletal muscle, triphasic WT	Vincristine and actinomycin-D	No recurrence (6 months)
Keskin [19]	2011	19 y	M	Testis	4.0 cm	A mature cystic teratoma with a focus of nephroblastomatous elements	Bleomycin, etoposide, and cisplatin (4 cycles). Second-line paclitaxel, ifosfamide, and cisplatin.	Death (18 months)
Ishida [20]	2012	2 m	F	Retroperitoneum	13.0 × 10.0 × 10.0 cm	Multilobulated nephroblastomatous elements within an immature teratoma	None	No recurrence (3 months)
Baskaran [21]	2013	3 y	M	Retroperitoneum	11.0 × 10.0 cm	A unilocular thin-walled cyst with one polypoid focus of a blastema in a mature cystic teratoma	None	No recurrence (1 year)
Ma [22]	2014	6 m	M	Sacrococcygeal region	9.0 × 8.0 × 5.0 cm	Mature heterologous tissue with one focus of nephroblastic elements	None	No recurrence (8 months)
Li [23]	2017	3 y	F	Abdomen	14.0 × 13.0 × 10.0 cm	A multifocal blastemal component in a mature cystic teratoma	None	No recurrence (2 years)
Unny [24]	2022	6 y	F	Retroperitoneum	9.2 × 8.0 × 5.2 cm	Initial biopsy was diagnosed as Ewing’s tumor. Resection after neoadjuvant chemotherapy revealed a mature cystic teratoma with glomeruloid structures and a focal blastema.	Salvage chemotherapy with vincristine, orinotecan, and temozolomide	Recurrence and metastasis (2 months)
Alfawaz (our case)	2024	8 y	F	Mediastinum	10.0 × 8.0 × 2.5 cm	Mature cystic teratoma with heterologous elements and small foci of nephroblastomatous elements	None	No recurrence (6 months)

Notes: RMS, rhabdomyosarcoma; WT, Wilms’ tumor; TWT, teratoid Wilms’ tumor.

**Table 3 ijms-25-12427-t003:** Early and late events.

Early Events	Late Events
1p Loss	
4q Loss	
7q Loss	7p LOH
11p Loss	
11p13 LOI and LOH	
12q Loss	16q LOH
22 Loss	
Mutations in *WT1*, *WTX*, and *KRAS*	Mutations in *CTNNB1* and *FBXW7*

Notes: numbers refer to chromosomes; p, the short arm of the chromosome; q, the long arm of the chromosome; LOH, loss of heterozygosis; LOI, loss of imprinting; *CTNNB1*, Catenin beta-1; *FBXW7*, F-box and WD repeat domain-containing 7, also known as *Sel10*, *hCDC4*, or *hAgo*; *KRAS*, Kirsten rat sarcoma virus; WT1, Wilms Tumor Suppressor Gene 1; *WTX*, Wilms Tumor Gene on X Chromosome (modified from Bánki et al. 2023 [56]).

## Data Availability

The original contributions presented in this case study are included in the article, further inquiries can be directed to the corresponding author (senior author).

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
