# Peer review of "Mediastinal Teratoma with Nephroblastomatous Elements: Case Report, Literature Review, and Comparison with Maturing Fetal Glomerulogenic Zone/Definitive Zone Ratio and Nephrogenic Rests"

_ijms, 2024, doi:10.3390/ijms252212427_

Round 1
Reviewer 1 Report
Comments and Suggestions for Authors
Mediastinal Teratoma with Nephroblastomatous Elements: Case Report, Literature Review, and Comparison with Maturing Fetal Glomerulogenic Zone / Definitive Zone Ratio and Nephrogenic Rests.
General comments
The article presents a case of extrarenal Wilms' tumour – a rare tumor entity usually seen as a mass in the retroperitoneal area. Starting from this case, the authors carry out a literature review and a comparative analysis with maturing fetal glomerulogenic zone/definitive zone ratio and nephrogenic rests.
The rarity of this neoplasm sustains the publication of this article.
In general, the paper is well written, but there are several issues that should be addressed.
Abstract
The abstract provides a clear, concise, and comprehensive summary of the article, including the case report and literature review. However, the authors do not mention anything about the comparison with maturing fetal glomerulogenic zone/definitive zone ratio and nephrogenic remnants - the idea of this comparative analysis should be included in the abstract.
Introduction
The number of cases of extrarenal Wilms' tumour documented in the literature differs between the abstract (less than 30 cases) and section Introduction (less than 100 cases). Reference 2 cites 72 papers with 98 pediatric patients (Karim A, Shaikhyzada K, Abulkhanova N, Altyn A, Ibraimov B, Nurgaliyev D, et al. Pediatric Extra-Renal Nephroblastoma 362 (Wilms’ Tumor): A Systematic Case-Based Review. Cancers. 2023 Apr 29;15(9):2563). The authors should use this number as a benchmark.
Case presentation
The current Section 2. Case presentation needs to be reorganized into subsections to allow tracking of case management from admission to death. In this organization, the pathological examination (currently Section 3. Results, with a single subsection – 3.1. Pathology) becomes a final subsection.
Confirmation of the diagnosis of TWT is based on histopathological examination. In this context, authors should complete the subsection Pathology with the following information
- number of blocks that were taken from the surgical resection piece (how many from the cystic structure and how many from the attached thymic tissue)
- characteristics of the antibodies used (in a table, with all information on clone, dilution, localization, manufacturer, tissue specificity)
- morphology of the attached thymic tissue
- differential diagnostic algorithm, supported by the positivity and negativity of the IHC markers (especially as the authors state in the abstract: This case underscores the importance of histopathological and immunohistochemical analysis in diagnosing TWT and differentiating it from other mediastinal tumors…..)
The legend of Figure 2 should be modified, removing the information on antibodies which does not provide anything relevant (i.e. WT1 expression using a monoclonal antibody against the Wilms tumor 1 should become anti-WT1 antibody) and the repetitive information on scale bars.
Discussion (section 4) / Narrative Literature Review (section 5) / Glomerulogenic Zone in the Ontogenesis (section 6) / Nephrogenic Rests (section 7)
These sections should be reorganized, in continuity with Section 2. Case presentation. We recommend that the Discussion becomes Section 3, including as subsections: 3.1. Narrative Literature Review, 3.2. Glomerulogenic Zone in the Ontogenesis, 3.3. Nephrogenic Rests.
The authors provide a comprehensive review of 23 cases, alongside the above case. The clinical details, site, histology, chemo-Rx treatment, and follow-up of the cases were clearly presented.
The analysis of maturing fetal glomerulogenic zone and nephrogenic rests represent a novel approach that has been well explored. However, the authors should better explain the relevance of the comparison with maturing fetal glomerulogenic zone / definitive zone ratio and nephrogenic rests, taking in consideration the relationship between these structures.
Section Discussion should be completed with a distinct subsection emphasizing the importance of histopathological and immunohistochemical analysis in the diagnosis of TWT and its differentiation from other mediastinal tumors (as stated in the abstract).

Author Response
Reviewer 1:
Thank you for your valuable feedback and insightful comments on our manuscript.
Comment: Abstract
The abstract provides a clear, concise, and comprehensive summary of the article, including the case report and literature review. However, the authors do not mention anything about the comparison with maturing fetal glomerulogenic zone/definitive zone ratio and nephrogenic remnants - the idea of this comparative analysis should be included in the abstract.
Response: Abstract
An additional statement was added to address that aspect of the paper in lines 26-28: “Our paper also provides an in-depth look at nephron development and nephrogenic rests, highlighting the structural and functional aspects of nephrogenesis and the factors that disrupt it in fetal kidneys.”
Comment: Introduction
The number of cases of extrarenal Wilms' tumour documented in the literature differs between the abstract (less than 30 cases) and section Introduction (less than 100 cases). Reference 2 cites 72 papers with 98 pediatric patients (Karim A, Shaikhyzada K, Abulkhanova N, Altyn A, Ibraimov B, Nurgaliyev D, et al. Pediatric Extra-Renal Nephroblastoma 362 (Wilms’ Tumor): A Systematic Case-Based Review. Cancers. 2023 Apr 29;15(9):2563). The authors should use this number as a benchmark.
Response: Introduction
Our paper focuses explicitly on extra-renal teratoid Wilms tumor (TWT), whereas the referenced article broadly addresses extra-renal Wilms tumor (EWT). The article in question describes 98 pediatric cases of Wilms tumor with classic morphology occurring outside the kidney, the typical site for these tumors.
In contrast, we limit our discussion to cases identified as teratoid Wilms tumor (TWT), a rare subtype characterized by at least 50% mixed teratomatous components and immature renal elements. Various terms have been used to describe this entity, including “teratoma with nephroblastoma,” “teratoma with nephroblastomatous elements,” and “teratoma with immature renal elements.”
While TWTs are infrequently reported within the kidney, extrarenal occurrences are exceedingly rare, and this unique presentation is the central focus of our paper.
Comment: Case presentation
The current Section 2. Case presentation needs to be reorganized into subsections to allow tracking of case management from admission to death. In this organization, the pathological examination (currently Section 3. Results, with a single subsection – 3.1. Pathology) becomes a final subsection.
Confirmation of the diagnosis of TWT is based on histopathological examination. In this context, authors should complete the subsection Pathology with the following information
- number of blocks that were taken from the surgical resection piece (how many from the cystic structure and how many from the attached thymic tissue)
- characteristics of the antibodies used (in a table, with all information on clone, dilution, localization, manufacturer, tissue specificity)
- morphology of the attached thymic tissue
- differential diagnostic algorithm, supported by the positivity and negativity of the IHC markers (especially as the authors state in the abstract: This case underscores the importance of histopathological and immunohistochemical analysis in diagnosing TWT and differentiating it from other mediastinal tumors…..)
The legend of Figure 2 should be modified, removing the information on antibodies which does not provide anything relevant (i.e. WT1 expression using a monoclonal antibody against the Wilms tumor 1 should become anti-WT1 antibody) and the repetitive information on scale bars.
Response: Case Presentation
The number of blocks was added in lines 83 and 84.
The morphology of the attached thymic tissue was added in lines 92 and 93.
Differential diagnosis and final diagnosis were added in lines 94-97.
The characteristics of the antibodies will be included in the appendix section.
The legend of Figure 2 was modified as advised in lines 114-116.
The section headings and subheadings were reorganized as advised.
Comment: Discussion (section 4) / Narrative Literature Review (section 5) / Glomerulogenic Zone in the Ontogenesis (section 6) / Nephrogenic Rests (section 7)
These sections should be reorganized, in continuity with Section 2. Case presentation. We recommend that the Discussion becomes Section 3, including as subsections: 3.1. Narrative Literature Review, 3.2. Glomerulogenic Zone in the Ontogenesis, 3.3. Nephrogenic Rests.
The authors provide a comprehensive review of 23 cases, alongside the above case. The clinical details, site, histology, chemo-Rx treatment, and follow-up of the cases were clearly presented.
The analysis of maturing fetal glomerulogenic zone and nephrogenic rests represent a novel approach that has been well explored. However, the authors should better explain the relevance of the comparison with maturing fetal glomerulogenic zone / definitive zone ratio and nephrogenic rests, taking in consideration the relationship between these structures.
Section Discussion should be completed with a distinct subsection emphasizing the importance of histopathological and immunohistochemical analysis in the diagnosis of TWT and its differentiation from other mediastinal tumors (as stated in the abstract).
Response: Discussion (section 4) / Narrative Literature Review (section 5) / Glomerulogenic Zone in the Ontogenesis (section 6) / Nephrogenic Rests (section 7)
Section headings and subheadings were reorganized as advised.
The nexus glomerulogenic zone and nephrogenic rests (NRs) and Wilms tumor has been extensively examined and elucidated based on our present understanding. Future research will be necessary, and laser capture microdissection is likely to play a crucial role soon. As indicated above, NRs are significantly persistent clusters of embryonic cells, indicative of microscopic abnormalities (dysplasias) in the developing kidney. Although NRs are primarily recognized as precursors to Wilms tumor, numerous alternate outcomes are evident, with the majority of rests ultimately leading to atresia. NRs exhibit a distinct boundary with the surrounding kidney and do not possess a pseudocapsule, while Wilms tumors consistently display a pseudocapsule that separates the tumor from the underlying kidney. Nephroblastomatosis refers to multifocal or diffuse nephrogenic rests, indicating a more widespread disease state. Universal nephroblastomatosis signifies the whole substitution of the renal lobe by nephrogenic tissue. The outcomes of nephrogenic rests and nephroblastomatosis include regression, sclerosis, quiescence, hyperplasia, or neoplasia. Evidence strongly indicates that the neoplastic transformation of nephrogenic rests leads to nephroblastoma. NRs predominantly arise in conjunction with Wilms' tumor; perilobar rests exhibit a significant correlation with synchronous bilateral Wilms' tumors, while intralobar rests are more closely linked to metachronous tumors. Genetic research indicates that NRs frequently exhibit numerous chromosomal abnormalities similar to those found in Wilms' tumor, hence reinforcing the notion that they are progenitors to nephroblastoma. Consequently, NRs are acknowledged as therapeutically important entities need thorough diagnosis and vigilant monitoring. Increased awareness of the clinical significance of NRs and nephroblastomatosis has resulted in enhanced detection of these precancerous lesions, stimulated more rigorous research into their biological behavior, and prompted extensive discussions regarding potential new treatment protocols. The pathologists' capacity to identify and detect NRs has improved due to the enhanced Beckwith categorization (Hennigar RA, O'Shea PA, Grattan-Smith JD. Clinicopathologic features of nephrogenic rests and nephroblastomatosis. Adv Anat Pathol. 2001 Sep;8(5):276-89. doi: 10.1097/00125480-200109000-00005. PMID: 11556536.). Radiologists have utilized high-resolution imaging methods to enhance the detection of NRs in situ. Clinicians and surgeons are more cognizant of the influence that NRs exert on patient treatment. Notwithstanding this advancement, additional evidence is required to more precisely characterize these lesions. A NR that may subsequently evolve into Wilms tumor could emerge at any phase of glomerulogenic zone differentiation. However, latest insights into kidney development indicate that these rests likely originate from the initial stages of embryonic or fetal development, when the glomerulogenic zone is more pronounced compared to later stages of gestation. A comprehensive review of all molecular research on NRs conducted from 1990 to 2022. Bánki et al. (Bánki T, Drost J, van den Heuvel-Eibrink MM, Mavinkurve-Groothuis AMC, de Krijger RR. Somatic, Genetic and Epigenetic Changes in Nephrogenic Rests and Their Role in the Transformation to Wilms Tumors, a Systematic Review. Cancers (Basel). 2023 Feb 21;15(5):1363. doi: 10.3390/cancers15051363. PMID: 36900155; PMCID: PMC10000075.) have provided a comprehensive review of all molecular investigations on nephrogenic rests conducted between 1990 and 2022. Consequently, an extensive array of approaches was employed to investigate chromosomal areas, copy number variations, specific genes, and epigenetic alterations in NRs. Twenty-three papers were identified demonstrating the loss of chromosomal arms 11p13, 11p15, 1p, 4q, and 11p, alongside gains in 1q, 7q, and 12q, in addition to alterations in WT1 (Table 3).
Table 3. Early and Late Events
- 1p Loss
- 4q Loss
- 7q Loss 7p LOH
- 11p Loss
- 11p13 LOI and LOH
- 12q Loss 16q LOH
- 22 Loss
- Mutations in WT1, WTX, and KRAS Mutations in CTNNB1 and FBXW7
Notes: Numbers refer to chromosomes; p, short arm of the chromosome; q, long arm of the chromosome; LOH, loss of heterozygosis; LOI, loss of imprinting; CTNNB1, Catenin beta-1, FBXW7, F-box and WD repeat domain containing 7 (FBXW7), also known as Sel10, hCDC4 or hAgo; KRAS, Kirsten rat sarcoma virus; WT1, Wilms Tumor Suppressor Gene 1; WTX, Wilms Tumor Gene on X Chromosome.
Consequently, significant insights have been gained into the genomic alterations in NR. Loss of heterozygosity at 11p13 and 11p15, along with the expression of IGF-2 and mutations in WT1 and WTX, seem to contribute to the early carcinogenesis of Wilms tumor. Loss of heterozygosity at 16q and 7p, along with mutations in CTNNB1, appear to manifest later in development. The methylation patterns of NR are comparable to those of WT. Owing to swift advancements in genome-wide molecular techniques, enhanced utilization of formalin-fixed and paraffin-embedded material, and the accessibility of histologically validated frozen specimens, genetic alterations in NRs and corresponding Wilms tumor can be examined in larger cohorts, potentially elucidating the initial stages of tumorigenesis of the nephroblastoma. Single cell transcriptomics associated with laser capture microdissection will probably key in the nearest future.

Reviewer 2 Report
Comments and Suggestions for Authors
The manuscript is well-written. I think your paper will be a useful resource for doctors dealing with similar situations. It also makes a significant contribution to the expanding body of knowledge on extrarenal TWT, which I appreciate.
Author Response
Reviewer 2
Comment:
The manuscript is well-written. I think your paper will be a useful resource for doctors dealing with similar situations. It also makes a significant contribution to the expanding body of knowledge on extrarenal TWT, which I appreciate.
Response
Thank you for your positive feedback. We are glad you find our manuscript well-written and a valuable resource for clinicians managing extrarenal TWT cases. We appreciate your recognition of our contribution to this field.
Thank you again for your time and valuable input.

Reviewer 3 Report
Comments and Suggestions for Authors
Dear Authors
The case is very interesting. I have some comments and questions.
1. Introduction
-extrarenal Wilms tumor and location in the mediastinum - how often?
-what is difference between extrarenal Wilms tumor (a teratoid form) and teratoma with nephroblastomatosis elements?
2. Case presentation
Surgery - did the tumor have capsula?
Stage?
Was fluid in the pleura before spilling within the right thorax and mediastinum during dissection?
Was the the fluid in the pericardium? Did the patient have cardiac tomponade? Only compression of the right antrum?
3. Results
where in the tumor was this one foci spanning 1.3 cm with atypical epithelial tissue with blastemal differentiation?
4. Discussion
Review
-what literature bases did you use to prepare your review?
-what were key words which you use to do it?
-who was treated by chemo- and radiotherapy? the indications for chemo- and radiotherapy? What kind of chemotherapy was used?
-Is teratoid Wilms tumor classified as LR or SR or HR according to SIOP RTSG classification?
-Table 1 should include stages of a disease.
Generally the discussion is too long and requires shortening, especially subsections "Glomerulogenic Zone in the Ontogenesis" and "Nephrogenic rests"
Author Response
Reviewer 3
Thank you for your valuable feedback and insightful comments on our manuscript, which have been instrumental in enhancing its quality.
Comment:1. Introduction
-extrarenal Wilms tumor and location in the mediastinum - how often?
-what is difference between extrarenal Wilms tumor (a teratoid form) and teratoma with nephroblastomatosis elements?
Response: 1. Introduction
While broader discussions generally address extrarenal Wilms tumor (EWT), our focus is on extrarenal teratoid Wilms tumor (TWT).
EWT refers to tumors with classic WT morphology outside the kidney. In contrast, we concentrate on cases specifically characterized as teratoid Wilms tumor (TWT), a rare subtype variably defined by at least 50% mixed teratomatous components combined with a component of immature renal elements.
Although TWTs are rarely reported in the kidney, extrarenal occurrences are exceedingly uncommon, making this unique presentation the central focus of our analysis.
Because of the non-standardized nomenclature, various terms have been used to describe similar entities, including “teratoid wilms tumor”, “teratoma with nephroblastoma,” “teratoma with nephroblastomatous elements,” and “teratoma with immature renal elements.”
Of the extrarenal TWT, 12.5% (3 cases, including our case) occurred in the mediastinum
Comment:1. Surgery
Did the tumor have capsula?
Stage?
Was fluid in the pleura before spilling within the right thorax and mediastinum during dissection?
Was the the fluid in the pericardium? Did the patient have cardiac tomponade? Only compression of the right antrum?
Response: Surgery
The tumor had a capsule and was intact at the time of operation. No fluid was documented in the pleura prior to dissection. The intact description was added in line 74.
There is no current staging system with extrarenal teratoid wilms tumor. No neoadjuvant therapy was administered in our case. There was a reported intraoperative spillage of some cyst contents but the lesion was considered completely excised. No other lesions were identified by imaging.
There was no fluid in the pericardium. The lesion was adherent to the pericardium without invasion. Only compression of the right atrium was noted.
Comment: 3. Results
where in the tumor was this one foci spanning 1.3 cm with atypical epithelial tissue with blastemal differentiation
Response: 3. Results
Since the focus of immature elements was small and non-mass-forming, the precise location within the cystic teratomatous mass could not be determined accurately. Additional sections were submitted in an attempt to identify the location, but these were non-contributory.
Comment: 4. Discussion
Review
-what literature bases did you use to prepare your review?
-what were key words which you use to do it?
-who was treated by chemo- and radiotherapy? the indications for chemo- and radiotherapy? What kind of chemotherapy was used?
-Is teratoid Wilms tumor classified as LR or SR or HR according to SIOP RTSG classification?
-Table 1 should include stages of a disease.
Generally the discussion is too long and requires shortening, especially subsections "Glomerulogenic Zone in the Ontogenesis" and "Nephrogenic rests"
Response: 4. Discussion
Literature review for cases of teratoid wilms tumor was through PubMed, Scopus and Cochrane.
Statement regarding key words was added in lines 114-116 “Search keywords used include “teratoid Wilms’ tumor”, “teratoma with nephroblastoma-tous elements”, “teratoma with Wilms’ tumor”, “teratoma with immature renal elements”, and “teratoma with nephroblastoma””.
All cases mentioning features of TWT were individually reviewed with emphasis on extrarenal location. Our paper included all identified cases of extrarenal TWT.
There is insufficient data to support a grading scheme for extrarenal TWT and immature teratomas in the mediastinum. Using criteria for grading ovarian teratomas, the presence of immature elements over a span of 3 low power fields is considered high grade. Conversely, using histologic stratification of Wilms tumor, the foci of immature renal elements identified in our lesion is predominantly epithelial with no evidence of anaplasia. Therefore, it may be considered low risk according to the international guidelines of pediatric oncology.
Due to variable staging info in the references articles and lack of standardized sraging criteria, stage was not included in Table 1.
We tried to delete irrelevant sentences, but the other two reviewers did not request to shorten the discussion, who asked conversely to expand the section on the nexus glomerulogenic zone and Wilms tumor.

Round 2
Reviewer 3 Report
Comments and Suggestions for Authors
Dear Authors
your article is very interesting.
I don't have additional questions.